# Ultrasensitive Detection of Interleukin 6 by Using Silicon Nanowire Field-Effect Transistors

**DOI:** 10.3390/s23020625

**Published:** 2023-01-05

**Authors:** Wen-Pin Hu, Yu-Ming Wu, Cao-An Vu, Wen-Yih Chen

**Affiliations:** 1Department of Bioinformatics and Medical Engineering, Asia University, Taichung 41354, Taiwan; 2Department of Chemical and Materials Engineering, National Central University, Taoyuan City 32001, Taiwan

**Keywords:** aptamer, COVID-19, field-effect transistors, interleukin 6, infectious diseases

## Abstract

Interleukin 6 (IL-6) has been regarded as a biomarker that can be applied as a predictor for the severity of COVID-19-infected patients. The IL-6 level also correlates well with respiratory dysfunction and mortality risk. In this work, three silanization approaches and two types of biorecognition elements were used on the silicon nanowire field-effect transistors (SiNW-FETs) to investigate and compare the sensing performance on the detection of IL-6. Experimental data revealed that the mixed-SAMs-modified silica surface could have superior surface morphology to APTES-modified and APS-modified silica surfaces. According to the data on detecting various concentrations of IL-6, the detection range of the aptamer-functionalized SiNW-FET was broader than that of the antibody-functionalized SiNW-FET. In addition, the lowest concentration of valid detection for the aptamer-functionalized SiNW-FET was 2.1 pg/mL, two orders of magnitude lower than the antibody-functionalized SiNW-FET. The detection range of the aptamer-functionalized SiNW-FET covered the concentration of IL-6, which could be used to predict fatal outcomes of COVID-19. The detection results in the buffer showed that the anti-IL-6 aptamer could produce better detection results on the SiNW-FETs, indicating its great opportunity in applications for sensing clinical samples.

## 1. Introduction

The outbreak of the COVID-19 pandemic started in late December 2019 and then spread quickly worldwide, leading to a global public health emergency. Quarantine measures were widely used in countries around the world to prevent the rapid spread of the disease. Many COVID-19 rapid test kits for the detection of antigens were developed to meet the need for rapid and large-scale screening to confirm whether someone was infected with the SARS-CoV-2 virus. Except for the COVID-19 rapid test kit, the real-time transcription-polymerase chain reaction (RT-PCR) is the gold standard for monitoring COVID-19 infection, which can provide qualitative and quantitative detections [1]. Silicon nanowire field-effect transistors (SiNW-FETs) have attracted great interest as biosensors because they provide high-sensitivity, label-free, and real-time detection and can be mass-produced in the semiconductor industry. A field-effect transistor (FET)–based biosensing device with a coating layer of graphene sheet on the sensing surface has been reported to be able to detect the SARS-CoV-2 spike protein at concentrations of 1 fg/mL [2]. In addition, the two-dimensional-material-based FET biosensor is reported to have a detection limit of 25 pg/mL in 0.01X phosphate-buffered saline (PBS) for detecting the SARS-CoV-2 spike protein by using a monoclonal antibody [3].

In the clinical classification of COVID-19, patients can be classified into three types: mild, common, and severe critical [4]. Some elevated cytokine levels, such as interleukin-6 (IL-6), tumor necrosis factor (TNF), interferons (IFNs), granulocyte-macrophage colony-stimulating factor (GM-CSF), and interleukin-18 (IL-18), are often reported in severe-critical patients with COVID-19. The severity of COVID-19 is linked to cytokine storm, which is the surge in inflammatory molecules following COVID-19 infection [5,6]. Among these cytokines, IL-6 plays a vital role in human immune regulation, hematopoiesis, and inflammation. IL-6 consists of 184 amino acids with a molecular weight of 21 kDa, having a four-helix bundle structure. In healthy populations, IL-6 levels are thought to be undetectable or below 10 pg/mL [7,8]. However, the overexpression of IL-6 can lead to respiratory dysfunction, poor outcomes, and mortality [9] in COVID-19-infected patients. The level of IL-6 is a good predictor of the need for mechanical ventilation [10] or the identification of disease progression in severe or nonsevere conditions [11,12].

Data from 89 hospitalized COVID-19-infected patients are analyzed, and this study finds that the optimal cutoff for the maximal IL-6 level is 80 pg/mL for the prediction of respiratory failure [10]. In other studies, the severity of COVID-19 can be predicted by using the optimal threshold of IL-6 over 24.3 pg/mL [13] or 26.09 pg/mL [14]. In total 48 patients are classified as the mild group, and the average level of IL-6 level expressed in this group is 10.6 pg/mL, ranging from 5.13 pg/mL to 24.18 pg/mL [13]. Even though the cutoff value for the IL-6 level appears to vary from study to study, these studies have all shown that the monitoring of IL-6 levels helps predict the severity of COVID-19-infected patients.

For gathering data on cytokine levels in patients, electro-chemiluminescent immunoassay (ECLIA), flow cytometry, enzyme-linked immunosorbent assay (ELISA), and others are commonly used biochemical assays. Aside from predicting the severity of COVID-19, IL-6 is also utilized as a biomarker to predict various diseases. Some research groups have developed several methods for detecting IL-6 on biosensors [15]. The single-walled carbon nanotube (SWCNT) array containing deposited gold nanoparticles has been proposed by Yang et al. [16] to detect IL-6 in serum samples with a detection limit as low as 0.01 fg/mL, having a linear response range of the sensor from 0.01 to 100 fg/mL. This SWCNT array is an ultrasensitive platform for detecting IL-6; however, the level of IL-6 in different pathological conditions is usually higher than the linear response range of the SWCNT/Au sensor [15], including that of COVID-19. Tertiş et al. [17] proposed the electrochemical aptasensor, based on gold and polypyrrole nanoparticles, which has a linear domain for IL-6 detection ranging from 1 pg/mL to 15 μg/mL and with a detection limit of 0.33 pg/mL. The IL-6 levels of patients with rheumatoid arthritis (RA) are measured with an interdigitated aptasensor, and the sensor can provide a linear range from 0.021 pg/mL (1 fM) to 2100 pg/mL (100 pM) [18].

Herein, this work describes a SiNW-FET platform used for detecting IL-6 and investigated the influence of using different silanization methods and biorecognition elements on the performance of the sensor. The linear ranges of IL-6 detection by using the anti-IL-6 antibody and anti-IL-6 aptamer on the SiNW-FET platform were compared in this study. In addition, the feasibility of the sensor used in the application for predicting the severity of COVID-19-infected patients was evaluated. When the anti-IL-6 antibody was used as the recognition molecule, the detection ability of the SiNW-FET could distinguish only severe COVID-19-infected patients with a high-level expression of IL-6. Using the anti-IL-6 aptamer, the SiNW-FET provided a relatively wide linear range of IL-6 detection and could reach a lower concentration of valid detection down to 2.1 pg/mL (100 fM). The detection range for the proposed aptamer-functionalized SiNW-FET is enough to measure the IL-6 levels expressed in COVID-19-infected patients and to distinguish the mild and severe degrees of pathological conditions.

## 2. Materials and Methods

### 2.1. Reagents and Chemicals

Silane-polyethylene glycol-amine (silane-PEG-NH2; Cat. No: HE003005-1K) and silane-polyethylene glycol-hydroxy (silane-PEG-OH; Cat. No: HE003002-1K) were brought from Biochempeg Scientific Inc. (Watertown, MA, USA) for the formation of a self-assembled monolayer (SAM) on the device surface. Glutaraldehyde (GA; Product No.: 354400), bis-tris propane (BTP; Product No.: B6755), sodium cyanoborohydride (Product No.: 156159), 3-aminopropyltriethoxysilane (APTES; Product No.: 440140), disodium phosphate (Product No.: 106586), and monosodium phosphate (Product No.: S8282) were acquired from Sigma (Saint Louis, MO, USA). Tris (Product No.: J60202.K2) was purchased from USB Corporation (Cleveland, OH, USA). Acetone and ethanol (95%) were brought from Echo Chemical Co. (Miaoli County, Taiwan). We also acquired absolute ethanol (99.9%; EtOH) from JT Baker (Phillipsburg, NJ, USA). Mouse monoclonal antibody (Ab) targeted to IL-6 (Anti-IL-6 antibody; Product No.: ARG56591) was received from arigo Biolaboratories Corp. (Hsinchu City, Taiwan). Human IL-6 protein was obtained from R&D Systems Inc. (Minneapolis, MN, USA; Product No.: 206-IL). In addition, the anti-IL-6 aptamer (Apt) with a length of 31 bases was synthesized by MDBio, Inc. (New Taipei City, Taiwan). The sequence of the anti-IL-6 aptamer was: 5’-NH_2_-C_6_-GGTGGCAGGAGGACT- ATTTATTTGCTTTTCT-3’ [18,19]. Phosphate-buffered saline (PBS buffer; Cat.No: PT-BF205-1L) was brought from Protech Technology Enterprise Co., Ltd. (Taipei City, Taiwan). The pH values of the BTP and PBS buffers used in this study were adjusted to 7.4. The lab of Prof. Chun-Jen Huang at the Department of Chemical and Materials Engineering of the National Central University supplied the chemical of 1-(3-aminopropyl) silatrane (APS). Other chemicals used in this research were all reagent grade.

### 2.2. Instrumentation

The surface of the SiNW-FET device was cleaned using the oxygen plasma cleaner (Harrick Plasma, New York, NY, USA) before the surface modification. Each n-type SiNW-FET device had two nanowires, and each nanowire had a length of 2 μm and a width of 200 nm [20], measurements that had been used in previous reports [20,21,22,23]. The SiNW-FET measurements were carried out on the chip-on-board (COB) detection platform provided by Helios Bioelectronics, Inc. (Hsinchu County, Taiwan). Atomic force microscopy (AFM; SPA-400 DFM, Seiko, Japan) was utilized to analyze the surface roughness of the FET surface after each modification step. In addition, X-ray photoelectron spectroscopy (XPS; Thermo VG-Scientific, Loughborough, UK) was applied to identify the elemental composition and chemical state of the SiNW-FET surface.

### 2.3. Chemical Modification of the SiNW-FET Decive

The acquired SiNW-FET devices needed to be cleaned thoroughly by using, in order, ethanol and deionized (DI) water and then blown dried with nitrogen gas. Afterward, the SiNW-FET devices were placed in the oxygen plasma cleaner to operate at 18W for 10 min. After the treatment of oxygen plasma, the SiNW-FET devices were then cleaned with a rinse of ethanol and deionized (DI) water, in that order again, and blown dry with nitrogen gas. Next, the SiNW-FET devices were placed in an oven for 10 min at 120 °C to remove the residual water from the device surface. Finally, the oxygen plasma cleaner was used to treat the SiNW-FET device again to decompose the surface contaminants and form a new oxide layer with the OH groups for further chemical modifications. Three silanization methods were used to perform the subsequent surface modifications: (1) mixed-SAMs solution (mSAMs, silane-PEG(1k)-NH2 and silane-PEG(1k)-OH with a molar ratio of 1:10 in EtOH); (2) APTES solution (2% in EtOH); and (3) APS solution (5 mM in EtOH).

For the first method of chemical modification, the SiNW-FET device was immersed in a microtube containing the mixed-SAMs solution. After sealing the microtube, it was placed in a vacuum container to evacuate the air, and the microtube was shaken at 30 °C for 30 min. After the reaction, the SiNW-FET device was removed and rinsed with EtOH to remove the unbound silane-PEG molecules. Afterward, the device was placed on a heating plate at 120 °C for 10 min to form the Si-O-Si bonds between silane molecules. In addition, for the APTES functionalization, the SiNW-FET device was immersed in the APTES solution at 40 °C for 1 h. After that, the SiNW-FET device was rinsed thoroughly with EtOH, and the device was also heated at 120 °C for 10 min. For the APS functionalization, the SiNW-FET device was immersed in the APS solution after the oxygen plasma treatment to react at 80 °C for 14.5 h. The device was washed with PBS buffer 5 times after the reaction, rinsed once with DI water and alcohol, and dried with nitrogen. The final step of APS functionalization was to place the device in an oven at 100 °C for 2 h.

### 2.4. Immobilization of Anti-IL-6 Antibody or Aptamer

Glutaraldehyde (2.5%) dissolved in 10 mM PBS buffer was used to react with the mixed SAMs, APTES-modified layer, or APS-modified layer on the SiNW-FETs at room temperature for 30 min in a dark container. As a result, the aldehyde groups (–CHO) of GA could covalently crosslink with the amine groups (–NH_2_) on the existing monolayer on the device surface, thereby forming a layer of an aldehyde group on the device surface for the next step of immobilizing biomolecules. To remove the unbound GA molecules, PBS buffer (10 mM, pH 7.4) was used to wash the SiNW-FET device, and then the device was rinsed with DI water.

To immobilize the antibody, the SiNW-FET device after the treatment of GA was immersed in the antibody solution (anti-IL-6 antibody (100 μg/mL) and 0.4% sodium cyanoborohydride in 1X PBS, pH 7.4) at 4 °C in a refrigerator overnight. The amine groups of antibodies could react with the aldehyde groups on the device surface to form covalent bonds. The SiNW-FET device was sequentially rinsed with 1X PBS and DI water, and the device was then immersed in the blocking solution (0.4% sodium cyanoborohydride in 10 mM tris-HCl, pH = 7.4) to shake for 30 min at room temperature. After treating sodium cyanoborohydride, the C = N groups on immobilized biomolecules could become more stable through reductive amination. At the same time, the tris molecule in tris-HCl could quench unreacted aldehydes to reduce the error signals produced by the nonspecific binding of the biomolecule. After rinsing the device with DI water, the SiNW-FET device with the immobilized anti-IL-6 antibodies on the surface was ready for use in the measurement.

For the immobilization of the anti-IL-6 aptamer, only the mixed-SAMs solution was adopted to carry out the silanization process for the surface of the SiNW-FET device. All procedures for silanization and the formation of the GA-modified layer were identical to the descriptions mentioned above. After the treatment of GA, the SiNW-FET device was immersed in the aptamer solution (anti-IL-6 aptamer (5 μM) and 0.4% sodium cyanoborohydride in 50 mM PBS buffer, pH 7.4) at room temperature for 1 h. At this step, the amine group modified at the 5’ end of the aptamer could react with the aldehyde group on the device surface to form the covalent bond. After rinsing with DI water, the SiNW-FET devices were also immersed in the blocking solution and shaken for 30 min at room temperature to reduce the nonspecific binding of biomolecules in FET measurements. Finally, the SiNW-FET devices were rinsed with DI water to complete the immobilization of aptamers.

### 2.5. SiNW-FET Measurements

The SiNW-FET device used in this study was wired and packaged on the printed circuit board (PCB) (shown in Appendix A). The packaged COB SiNW-FET contained 12 SiNW-FET devices (shown in Appendix A), and each SiNW-FET device had two nanowires. Therefore, the total number of nanowires in the packaged COB SiNW-FET was 24, and the signals from 12 channels were recorded in the experiments (Ch1 = device 1, Ch2 = device 2, and so forth; the signal was generated from 2 nanowires for each SiNW-FET device). The fluid channel was designed and fixed upon the SiNW-FET device to keep the liquid in the nanowire region. The electrical signals from 12 channels could be measured in each experiment by connecting the external electric contacts on the PCB. The FET measurement steps used in this study were similar to those in our previously published studies [22,23]. The steps were as follows: (1) The sensing buffer (10 mM BTP, pH 7.4) was injected into the fluid channel to obtain a stable I_d_–V_g_ curve as the baseline. (2) After this measurement, the sensing buffer was again injected into the fluid channel to measure the electrical signals as the data of the blank experiment. (3) The PBS buffer was injected into the fluid channel to wash the nanowire region. (4) The analyte solution (IL-6 in the PBS buffer, pH 7.4) was injected into the fluid channel and stood for 30 min. (5) After standing for 30 min, the PBS buffer was injected to wash off the analyte molecules that did not successfully bond with the anti-IL-6 antibody or anti-IL-6 aptamer. (6) Finally, the sensing buffer was again injected into the fluid channel to record the I_d_–V_g_ curve as the experimental data for acquiring the electric signals after the biorecognition event. After the measurement of the first concentration of IL-6 (2.1 pg/mL (100 fM)) was finished, Steps 3 to 6 were repeated to sequentially measure the other concentrations of IL-6 (21 pg/mL (1 pM) to 210 pg/mL (10 nM)). Three packaged COB SiNW-FETs were used in each experimental condition, and the experimental conditions were composed of modified layers and biorecognition elements. As was performed the previously published reports [22,23], the change in gate voltage before and after the biorecognition event was recorded, and the quantitative data were obtained by analyzing the voltage changes in the I_d_–V_g_ curves at a drain current (I_d_) of 1 × 10^−9^ A. In the measurement of I_d_–V_g_ curve, the drain (V_d_) and the source (V_s_) voltages were set at 1 and 0.5 volts, respectively. In addition, the gate voltage was swept from 0 V to 2 V with a sweep interval of 0.1 V. The stability of the SiNW-FETs without any modification had been tested in the BTP buffer (50 mM, pH7.4); the I_d_–V_g_ curves were measured 20 times, and the first I_d_–V_g_ curve for each channel was adopted as the baseline (shown in Appendix A). The values of the average shift and standard deviation of the gate voltage at the drain current (I_d_) of 1 × 10^−9^ A were 0.2 mV and 5.5 mV, respectively. The SPSS software (SPSS Inc., Version 18.0, Chicago, IL, USA) was used to perform the statistical analysis. The data obtained from detecting IL-6 with different concentrations on the devices with the mSAMs-GA-Apt layer were examined using a one-way analysis of variance (ANOVA) with Tukey’s HSD test.

### 2.6. Preparation of AFM and XPS Samples

The monitor wafers were used to prepare the AFM and XPS analysis specimens. The steps used for the chemical surface modification on the monitor (bare silicon) wafer were the same as for the SiNW-FET device. The steps applied in the immobilization of biomolecules, for antibody or aptamer, were also identical to those in the abovementioned section: immobilization of anti-IL-6 antibody or aptamer. Because silica surfaces of the monitor wafer and the SiNW-FET device have the same properties, the changes in the surface morphology or elemental composition of the monitor wafer could reflect the conditions on the surface of the SiNW-FET device.

## 3. Results

### 3.1. Morphology of Modified Silica Surfaces

The surface morphology of the chemically modified monitor wafer was analyzed using AFM. The surface conditions of the bare wafer and the wafers with different silanization methods are shown in Figure 1, and each silanization method has two representative AFM images. The areas of scanned AFM images are 2 μm × 2 μm, and two individual scans for each silanization method are shown in Figure 1a–f. The average roughness (Ra) and root mean square (RMS) roughness values for the bare silicon wafer are 0.1002 nm and 0.1311 nm. From the AFM images acquired from APTES-, APS-, and mixed-SAMs-modified silica surfaces, it was worth noting that mixed-SAMs-modified surfaces had relatively smaller values of Ra and RMS roughness compared with the APTES- and APS-modified surfaces. These results indicated that the mixed-SAMs layer could make the surface chemical grafting more uniform. Hence, the mixed-SAMs solution was used to modify the surfaces of SiNW-FET sensors for immobilizing the antibody or aptamer in the next step.

### 3.2. Morphology and Characteristics of the Device Surfaces with Antibodies or Aptamers

According to AFM measurements, the mixed SAMs had a small Ra value of 0.2 nm, which implied it could form the most uniform layer on the silica surface. Then, GA was adopted to treat with the mixed-SAMs-modified silica surface for the following immobilization of antibody or aptamer. The morphology and characteristics of the silica surfaces with the immobilized antibodies or aptamers were examined by using AFM and XPS analyses. Figure 2 shows the AFM images of wafer surfaces with immobilized antibodies and aptamers. Theoretically, the anti-IL-6 antibody had a molecular weight of about 150 kDa, standing upright on the surface, with a length of about 8.4 nm [24].

Meanwhile, the anti-IL-6 aptamer used in this study has 31 nucleic acids with a molecular weight of 9757.3 Da that is calculated by using OligoCal [25]. According to information from line profiles of AFM images (Figure 2a,b), the height of the immobilized antibody was around 10 nm, which was slightly higher than the theoretical value. The line profiles of the AFM images (Figure 2c,d) for the silicon wafer surfaces with immobilized aptamers indicated that the average height was around 9 nm. The sizes of aptamers are usually about 3–5 nm [26]. However, the Mfold web server [27] predicts the anti-IL-6 aptamer can have two structures (shown in Appendix A). Because each base pair has a length of 0.34 nm, one of the predicted structures can have a length of up to 8.16 nm. On the basis of the AFM results, we speculated that the structure shown in Appendix A might be the aptamer structure presented on the sensor surface. The measured height of the immobilized aptamer was also slightly higher than the expected value.

Moreover, the XPS survey spectra were applied to evaluate the elemental and atomic compositions on the wafer surface. Figure 3 shows the XPS survey spectra of the bare wafer, the wafer surface with immobilized anti-IL-6 antibody, and the wafer surface with immobilized anti-IL-6 aptamer. The survey spectra of the bare wafer showed the presence of major peaks due to elements such as oxygen (O 1 s) and carbon (C 1 s) peaks. Upon the immobilization of the anti-IL-6 antibody, a significant increase in carbon (from 17.6 to 43.6%) and nitrogen (N 1 s) (from 0.3% to 2.1%) was observed. In addition, a significant increase in carbon (from 17.6% to 28.6%) and nitrogen (from 0.3% to 1.5%) was also observed after the immobilization of the anti-IL-6 aptamer. The increase in N 1 s intensity at the Si-mSAMs-GA-Ab surface was attributed to the peptide bonds within the immobilized antibody. On the other hand, the nitrogen atom within the aptamer base was the main reason for the increased intensity of N 1 s at the Si-mSAMs-GA-Apt surface. In addition, the increase in the peak of C 1 s was mainly contributed by the mSAMs composed of the PEGs with long carbon chains in the structures. Figure 4 shows the high-resolution spectra of N 1 s for bare silicon wafer, Si-mSAMs-GA-Ab surface, and Si-mSAMs-GA-Apt surface. For the Si-mSAMs-GA-Ab surface (Figure 4b), the binding energies were at 402.1, 400.1, and 399.0 eV, corresponding to C-NH-C, N-H, and N-H_2_, respectively. The deconvolution of the N 1 s for Si-mSAMs-GA-Apt surface in Figure 4c was fitted into three components. The components at 402.6, 401.3, and 399.9 eV were assigned to C-N-C, N-H, and N-H_2_, respectively. According to the AFM and XPS analyses, the anti-IL-6 antibody or anti-IL-6 aptamer was confirmed to be successfully immobilized on the wafer surface.

### 3.3. Detection of IL-6 with the Antibody

To investigate the differences in the measurement of IL-6 using different silanization methods for the immobilization of antibodies, five concentrations of IL-6, ranging from 21 pg/mL (1 pM) to 210 ng/mL (10 nM), were used in the FET measurements. The quantitative data of gate voltage change and the calibration curve could be obtained. The I_d_–V_g_ curves were recorded in triplicate at room temperature (25 °C) to guarantee obtaining a stable electrical signal. All experiments were conducted using 10 mM BTP, and the values of gate voltage change in blank experiments were taken as the background noises. The gate voltage changes of the experiments must be three times that of the standard deviation of the blank experiments, implying that these values are effective signals [23]. The threshold of voltage change could be denoted as ΔV_th_. If the value of gate voltage change was below the value of ΔV_th_, it was interpreted as background noise. For the FET devices modified with APTES-GA-Ab, the standard deviation of the blank experiments was −9.71 mV. Therefore, the absolute value of the measured electrical signal must be greater than 29.13 mV to be regarded as a valid signal. The experimental data for detecting IL-6 for the FET devices modified with APTES-GA-Ab are shown in Figure 5a. From this figure, we could notice that the absolute values of voltage changes produced by the minimum (21 pg/mL, 1 pM) and the maximum (210 ng/mL, 10 nM) concentrations of IL-6 were greater than 29.13 mV, which were regarded as effective signals. For the other three concentrations, the absolute values of voltage changes could all be considered as background noises. In addition, the electrical signals did not show a trend with increasing concentrations of IL-6.

Figure 5b shows the experimental results for detecting IL-6 by using the FET devices with the modification of the APS-GA-Ab layer. The standard deviation of the blank experiments was −8.82 mV. Therefore, the measured electrical signals could be regarded as effective signals in that the absolute signal values were greater than 26.46 mV. From Figure 5b, it could be noticed that a trend existed with increasing concentrations of IL-6. However, the voltage changes of five concentrations were almost all less than 26.46 mV, so they were still regarded as background noises.

Then, the SiNW-FET devices with the modification of the mSAMs-GA-Ab layer were utilized to measure IL-6, and the standard deviation of the blank experiments was 3.42 mV. Hence, the absolute value of voltage change must be greater than 10.26 mV to be regarded as a valid signal. On the basis of the results shown in Figure 5c, the voltage changes in the detection of IL-6 with a concentration of 1 pM were regarded as background noises. When the concentration of IL-6 was increased to 10 pM (210 pg/mL) or higher, the signals could be considered valid. The saturation concentration is approximately 1 nM because the average value of voltage change did not obviously increase as the concentration of IL-6 reached 10 nM (210 ng/mL). Five examples of raw data for the detection of IL-6 on the SiNW-FETs with APTES-GA-Ab, APS-GA-Ab, and mSAMs-GA-Ab layers were provided in Appendix A, respectively.

### 3.4. Detection of IL-6 with the Aptamer

Compared with the antibody, the aptamer has some advantages, such as high stability, longer shelf life, less batch-to-batch variation, ease of synthesis, and easy chemical modification [28,29]. Hence, in this study, the anti-IL-6 aptamer was applied to replace the role of the anti-IL-6 antibody as the biorecognition element. The standard deviation of the blank experiments was −4.19 mV while using the anti-IL-6 aptamer in the FET measurements. Therefore, the absolute value of gate voltage change must be greater than 12.57 mV to be regarded as a valid signal. Figure 6 presents the change in gate voltage versus IL-6 concentration. The I_d_–V_g_ curves of blank and testing samples measured in the 12 channels of the SiNW-FET are shown in Appendix A, and Appendix A shows the enlargement image of one channel. From the data shown in Figure 6, it could be seen that the change in gate voltage was regarded as an effective signal when the concentration of IL-6 was above 100 fM (2.1 pg/mL). The calibration curve presented an approximately linear trend as the concentration of IL-6 started from 100 fM to 10 nM (210 ng/mL). For two adjacent groups, the statistical results indicated that significant differences appeared in two places (100 fM versus 1 pM, and 100 pM versus 1 nM). The statistical results are shown in Appendix A. When comparing these results of using the anti-IL-6 antibody, using the anti-IL-6 aptamer for detecting the IL-6 concentration of 100 fM could be taken as a valid signal, which was two orders of magnitude lower than that of the anti-IL-6 antibody.

## 4. Discussion

Owing to the properties of APTES, the quality of the silica surface modified by the APTES solution with a purity of 99.7% could be influenced by many factors, such as temperature, solvent, concentration, reaction time, and moisture [30]. Structural irregularities of the APTES-derived layer could form on the surface. Individual silane molecules were incorporated into the layer by reacting/interacting with the functionalities, such as electrostatic attraction, hydrogen bonding, covalent bonding, etc., shown at the interface [28]. In practice, the polycondensation product, physical adsorption, and molecular reversion were often generated in the APTES layer [31]. APS is a valuable alternative to APTES, which is less reactive and highly resistant to polymerization at neutral pH [32,33]. Concerning the surface preparation, the APS mica, compared with APTES mica, was better able to obtain reliable and reproducible AFM images for the mica surfaces with DNA and protein-DNA complexes in aqueous solutions [34]. For the SiNW-FET sensors, Debye screening length had limited the applications in detecting samples relevant to physiological conditions. Some studies showed that the sensors with PEG modification had significantly improved sensor responses by a threefold signal increase [35] and opened up opportunities for FET devices for biological sensing in physiological environments [35,36]. The AFM results in our study could prove that the mixed-SAMs-modified silica surface could have the smallest values of Ra and RMS in the three silanization methods. Therefore, the mixed-SAMs solution was adopted to modify the surfaces of SiNW-FET sensors. As for the APTES- and APS-modified surfaces, the results of AFM images were consistent with previous reports that structural irregularities commonly existed in the APTES-derived layer [30,31]. Hence, the values of the Ra and RMS roughness for APTES-modified surfaces are larger than those prepared with the other two silanization methods.

A comparison of the experimental data obtained by using the FET devices with different modification layers reveals that the devices with the modification of the mSAMs-GA-Ab layer exhibited the best performance. For the devices modified with the other two silanization methods, no significant trend was observed with the use of the APTES-GA-Ab layer, and the values of voltage change measured on the devices with the modification of APS-GA-Ab were all regarded as background noises. AFM measurements proved that the mixed-SAMs-modified silica had a more uniform surface than the APTES-modified and APS-modified silica surfaces after the chemical grafting. A more uniform surface of chemical grafting was beneficial for the subsequent immobilization of biomolecules. Previous studies have revealed that the PEG modification could locally reduce the effect of Debye screening, perhaps by lowering the relative permittivity compared with water [34,35]. In the analytical model, the Donnan potential picture was applied to explain the PEG effect, and the combination of recognition molecules, linker molecules, and PEG was supposed to make up a surface layer with practically constant potential [35]. Therefore, the Debye screening started to arise more away from the sensor surface, generating a higher signal if the analyte bound within the PEG layer [35]. With the benefit of increasing the effective screening length, the PEG layer on the FET could detect biomolecules even in high ionic strength solutions in real time [36,37].

The mSAMs-GA-Apt FET sensor used in this study could have valid results in detecting a concentration of IL-6 as low as 2.1 pg/mL (100 fM). According to the study reported by Gao’s group [13], clinical laboratory data with an IL-6 level over 24.3 pg/mL could predict the severity of COVID-19 with a sensitivity and specificity of 73.3% and 89.3%, respectively. In our study, the detection range of the mSAMs-GA-Apt FET sensor covered the concentration of IL-6 used to distinguish the patients with severe COVID-19. In this study, the anti-IL-6 aptamer showed advantages over the anti-IL-6 antibody in the FET measurements. We suggested that the aptamer had negatively charged phosphate groups, and the charge distribution was more uniform on the sensor surface. Furthermore, when comparing Figure 2b,d, the AFM images of the silicon wafer revealed that the number of immobilized aptamers was more than that of immobilized antibodies. Thanks to these properties, we supposed that this was why the detection of IL-6 could reach lower levels and that a wide detection range could be obtained when aptamers were used as probes.

However, the large values of standard deviations shown in Figure 6 might cause measurement uncertainties. The statistical results (Appendix A) revealed that the measured signals obtained from the group with IL-6 concentrations of 1 pM (21 pg/mL) could not have significant differences from its adjacent groups (10 pM and 100 pM). Therefore, using the SiNW-FETs with the mSAMs-GA-Apt layer for distinguishing the IL-6 concentrations in the range from 1 pM to 100 pM could be problematic.

## 5. Conclusions

In this work, the anti-IL-6 antibody or anti-IL-6 aptamer probe was applied as the biorecognition element on the SiNW-FETs to detect IL-6 and examine the ability of the SiNW-FETs to sense the target molecule. The silanization method of mixed SAMs could produce a uniform PEG layer that was shown by using AFM measurements. In addition, compared with the other two silanization methods, the electrical signals for the SiNW-FETs with the anti-IL-6 antibody immobilized upon the mixed SAMs were more prominent, and signals showed an approximately linear trend in increasing the concentrations of IL-6. When the anti-IL-6 antibody and anti-IL-6 aptamer were adopted in detecting IL-6, the results revealed that the detection range of the SiNW-FETs with the anti-IL-6 aptamer was broader (100 fM−1 nM) than that of the devices with the anti-IL-6 antibody. The valid detection of IL-6 by using an aptamer could reach a concentration as low as 100 fM, which was two orders of magnitude lower than using the anti-IL-6 antibody on the SiNW-FETs. Hence, the aptamer-functionalized SiNW-FETs may have great potential to be applied in identifying IL-6 levels in patients with severe COVID-19. However, in this study, the clinical samples from human tissue or serum were not measured on the aptamer-functionalized SiNW-FETs. Hence, further studies will be needed to verify that the aptamer-functionalized SiNW-FETs can meet the clinical needs for determining the severity of patients with COVID-19.

## Figures and Tables

**Figure 1 sensors-23-00625-f001:**
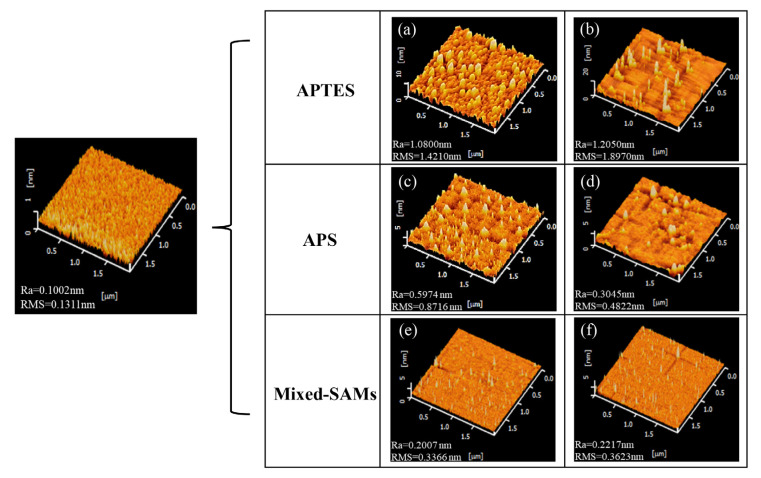
AFM images of silica surfaces. This figure shows two representative AFM images for each modified surface sampled in different areas. (**a**,**b**) APTES-modified silica surface; (**c**,**d**) APS-modified silica surface; and (**e**,**f**) mixed-SAMs-modified silica surface.

**Figure 2 sensors-23-00625-f002:**
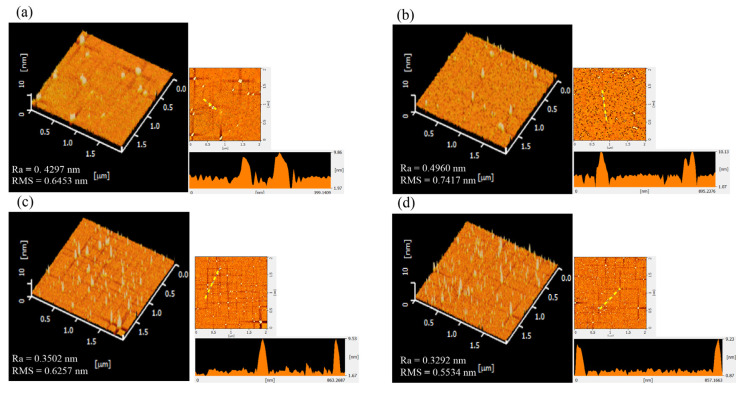
AFM images of silicon wafer surfaces with immobilized antibodies or aptamers. The device surfaces with immobilized antibodies (**a**,**b**) and aptamers (**c**,**d**). The 2D AFM image and line profile of height along the yellow line (the rectangular graph) are shown on the right side of each AFM image.

**Figure 3 sensors-23-00625-f003:**
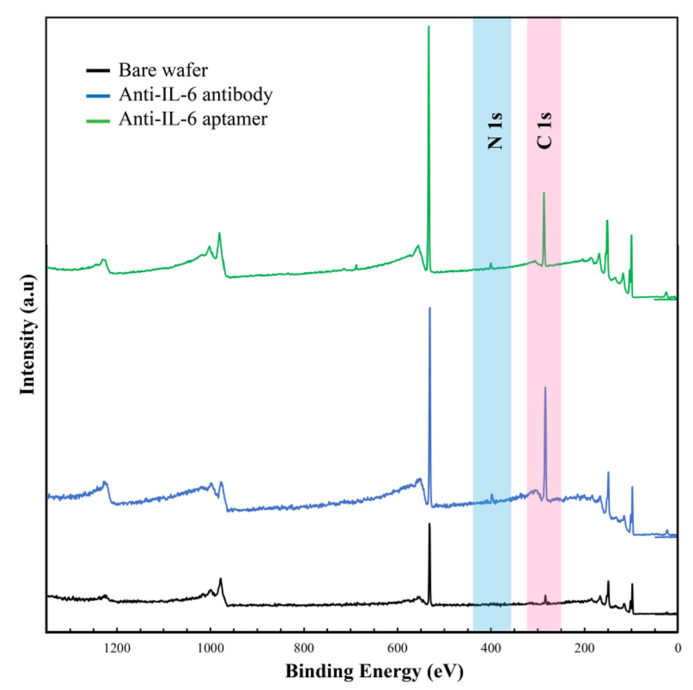
The XPS survey spectra of the bare wafer (in black), wafer surface with immobilized anti-IL-6 antibody (in blue), and wafer surface with immobilized anti-IL-6 aptamer (in green).

**Figure 4 sensors-23-00625-f004:**
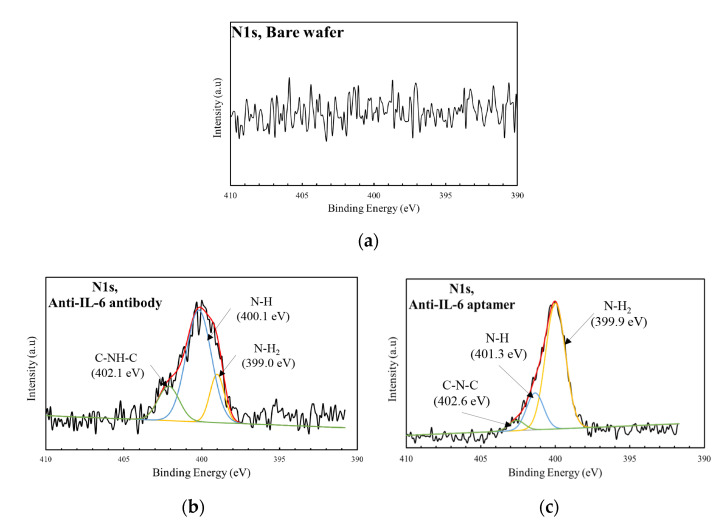
High-resolution spectra of N 1 s (**a**) bare silicon wafer, (**b**) Si-mSAMs-GA-Ab surface, and (**c**) Si-mSAMs-GA-Apt surface.

**Figure 5 sensors-23-00625-f005:**
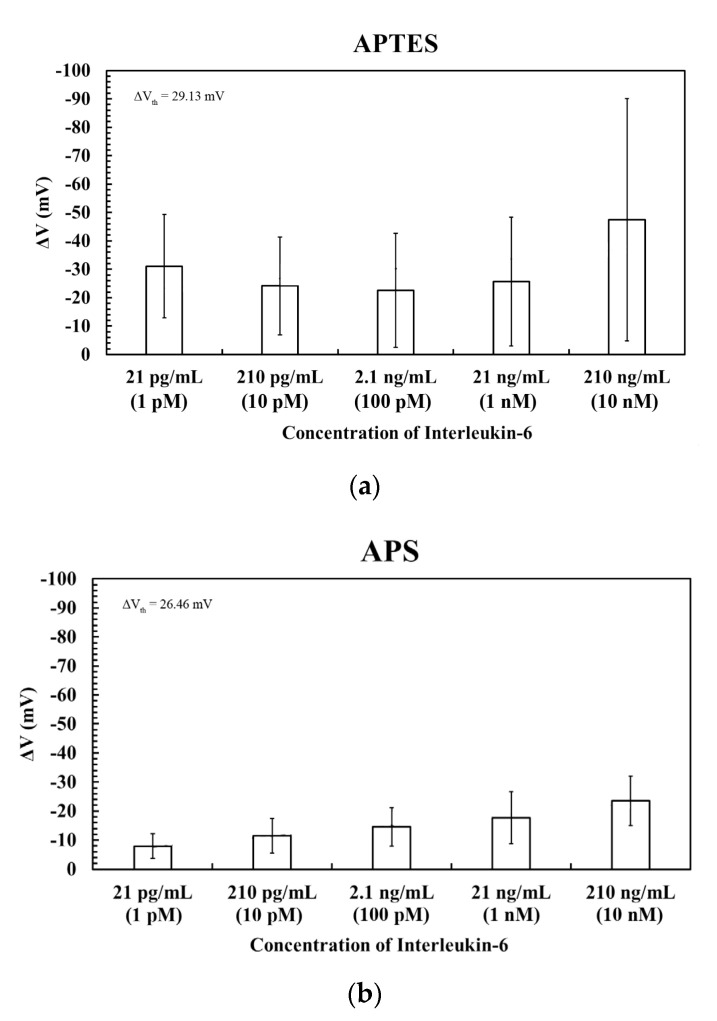
Detection of IL-6 on the SiNW-FETs by using the antibody. These experiments were performed on three modified layers upon the device surfaces: (**a**) APTES-GA-Ab layer, (**b**) APS-GA-Ab layer, and (**c**) mSAMs-GA-Ab layer. The threshold values, ΔV_th_, are denoted in the figures.

**Figure 6 sensors-23-00625-f006:**
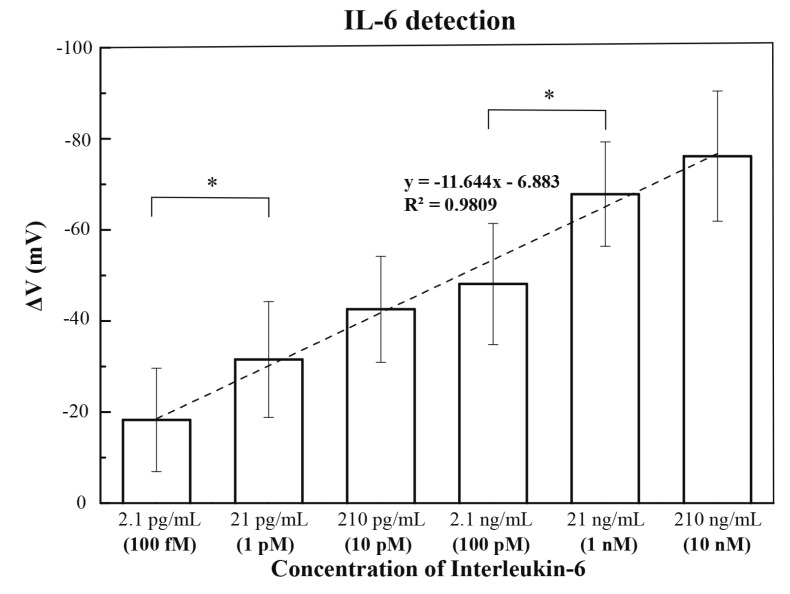
Detection of IL-6 on the SiNW-FETs with the mSAMs-GA-Apt layer. The average values of gate voltages and standard deviations for the 6 IL-6 concentrations from 100 fM to 10 nM are −18.2 ± 11.4, −31.5 ± 12.7, −42.6 ± 11.6, −48.1 ± 13.3, −67.9 ± 11.5, and −76.2, ± 14.3 mV, respectively. The asterisk symbols denote the significantly statistical difference between two adjacent groups (*p* < 0.05).

## Data Availability

The data presented in this study are available on request from the corresponding author.

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
