# Peer review of "Ultrasensitive Detection of Interleukin 6 by Using Silicon Nanowire Field-Effect Transistors"

_sensors, 2023, doi:10.3390/s23020625_

Round 1
Reviewer 1 Report
Summary:
This work presents the detection of interleukin 6 (IL-6) using silicon nanowire field-effect transistors (SiNW-FETs). To verify surface functionalization, three kinds of silane were tested on silicon wafers (instead of SiNW-FETs). The AFM results suggest that the SAM-based modification provided a smoother surface than the APTES- and APS-based ones did. Anti-IL-6 antibody and aptamer were respectively immobilized to measure electrical signals (i.e., ΔVg).
The authors announce that the limit of detection (LoD) of IL-6 measurements using aptamer is 100 fM, while the error bars are too remarkable to support the claim. To validate the detection, 1) the number of tested sensors should be revealed, 2) the measured data of blank samples should be presented, 3) the derivation of uncertainties (e.g., standard deviations) should be addressed, and, 4) most importantly, the measured results should statistically significant. In addition, information of the SiNW-FET fabrication, FET characteristics, several product numbers, and details of measurements are missing.
Overall, the missing data should be presented, detailed information of materials and methods are needed, and the data analysis should be significantly improved. As such, a major revision is expected.
In addition to the issues mentioned above, the reviewer’s comments can be found below.
Major comments:
1. Considering the presentation of data, several issues should be addressed:
1.a) To systematically compare the tested conditions, the measurements of aptamer using APTES and APS should be presented.
1.b) Please present the evidence of surface functionalization on “SiNW-FETs”
1.c) The time-lapse data of blank samples and testing samples should be provided.
1.d) Plotting and calculation of regression line on Fig.5 are seemingly meaningless, because i) the data of Fig.5a is obviously scattered, ii) uncertainties (error bars) of Fig. 5b and 5c are so remarkable, and iii) Langmuir isotherm or Henry’s equation might be better fitted with the data of Fig. 5c.
1.e) More terribly, the uncertainty shown in Fig. 6 is too significant to validate the detection.
2. Please provide characteristic data (e.g., Id-Vg and Id-Vd) of the sensor(s), or a reader cannot tell where the linear region of FET is.
3. To improve the credibility of this work, please consistently provide the product number of each materials/reagent.
4. What does the “theoretical” length of the aptamer come from? This is a bold argument (at Line. 241 and 242).
And the statement (at Line. 244) contradicts with the measured result (at Line. 238).
Minor comments:
1. The paper needs revising by a native English speaker. There are several awkward phrases and typos. Verb tense needs to be addressed and consistent.
2. In Introduction, the statements (at Line. 32 – 37 on Page.1) regarding COVID-19 testing methods should be revised. An antibody test is used to confirm the past infection, while an antigen test verifies current infection. And RT-PCR is still the gold standard when counting the number of COVID-19 cases.
Plus, there is a critical typo at Line. 90 on Page. 2.
Author Response
Response to Reviewers
This work presents the detection of interleukin 6 (IL-6) using silicon nanowire field-effect transistors (SiNW-FETs). To verify surface functionalization, three kinds of silane were tested on silicon wafers (instead of SiNW-FETs). The AFM results suggest that the SAM-based modification provided a smoother surface than the APTES- and APS-based ones did. Anti-IL-6 antibody and aptamer were respectively immobilized to measure electrical signals (i.e., ΔVg).
The authors announce that the limit of detection (LoD) of IL-6 measurements using aptamer is 100 fM, while the error bars are too remarkable to support the claim. To validate the detection, 1) the number of tested sensors should be revealed, 2) the measured data of blank samples should be presented, 3) the derivation of uncertainties (e.g., standard deviations) should be addressed, and, 4) most importantly, the measured results should statistically significant. In addition, information of the SiNW-FET fabrication, FET characteristics, several product numbers, and details of measurements are missing.
Response:
- The number of tested sensors is revealed in Section 2.5.
“The SiNW-FET device used in this study was wired and packaged on the printed circuit board (PCB) (shown in Figure S1(a)). In the packaged COB SiNW-FET, it contained 12 SiNW-FET devices (shown in Figure S1(b)), and each SiNW-FET device had two nanowires. Therefore, the total number of nanowires in the packaged COB SiNW-FET was 24, and the signals from 12 channels were recorded in the experiments (Ch1 = device 1, Ch2 = device 2, and so forth. The signal was generated from 2 nanowires for each SiNW-FET device.).”
“Three packaged COB SiNW-FETs were used in each experimental condition, and the experimental conditions were composed of modified layers and biorecognition elements. As the previously published reports [22,23], the change in gate voltage before and after the biorecognition event was recorded, and the quantitative data were obtained by analyzing the voltage changes in the Id−Vg curves at a drain current (Id) of 1 × 10−9 A.”
- We presented the measured data of blank samples on the SiNW-FET devices without any modification in Figure S2 of Supplementary Material. The values of average shift and standard deviation of gate voltage at the drain current (Id) of 1 × 10−9 A are 0.2 mV and 5.5 mV, respectively. Figures S5 and S6 also can give a clear picture about the measured data of blank samples.
- We added some sentences to address the derivation of uncertainties in the last paragraph of Discussion.
“However, the large values of standard deviations shown in Figure 6 might cause measurement uncertainties. The statistical results (Table S4) revealed that the measured signals obtained from the group with the IL-6 concentrations of 1 pM (21 pg/mL) could not have significant differences to its adjacent groups (10 pM and 100 pM). Therefore, using the SiNW-FETs with the mSAMs-GA-Apt layer for distinguishing the IL-6 concentrations in the range between 1 pM and 100 pM could be problematic.” - The statistical results by using one-way analysis of variance (ANOVA) with Tukey's HSD test are provided in Table S4. The statistical results reach statistical significance between adjacent groups are denoted in Figure 6.
Overall, the missing data should be presented, detailed information of materials and methods are needed, and the data analysis should be significantly improved. As such, a major revision is expected.
Response:
In this revised manuscript, we provide more data and detailed information of materials and methods. The statistical method is adopted to analyze the data.
In addition to the issues mentioned above, the reviewer’s comments can be found below.
Major comments:
- Considering the presentation of data, several issues should be addressed:
1.a) To systematically compare the tested conditions, the measurements of aptamer using APTES and APS should be presented.
Response:
Thanks for the reviewer's comment. I agree it is a good suggestion. The measurements of the FET devices prepared using APTES or APS and with the immobilization of aptamer may provide complete and systemic information. However, we didn't use APTES and APS to prepared the aptasensor in the measurements. The major reason was that the devices with the APTES-GA-Ab and APS-GA-Ab did not show good results compare with the results obtained from the devices with mSAMs-GA-Ab layer (as shown in Figure 5). For this reason, we gave up using APTES and APS in planning subsequent aptamer experiments. This is an excellent suggestion, and we think we will pay attention to the integrity of the experimental design when designing related experiments in the future.
1.b) Please present the evidence of surface functionalization on “SiNW-FETs”
Response:
I understand the reviewer's concern. However, in the design of this study, we didn't measure the electric properties of SiNW-FETs after each step of surface functionalization. The wafer surface should be the same as for the SiNW-FET device. We only adopted the AFM measurement to evidence surface functionalization. Besides, the XPS spectra can prove the aptamer and the antibody are immobilized on the sensor surface.
1.c) The time-lapse data of blank samples and testing samples should be provided.
Response:
Thanks for reviewer's suggestion. We provide the I-V curves of blank and testing samples in Supplementary materials. The SiNW-FET devices were packaged and installed into an electrical circuit board (as shown in Figure S1), and we could obtain signals from 12 channels (devices) (I-V curves as shown in the below Figure R1, named as Figure S4 in Supplementary Materials). Initially, the 10 mM BTP was injected to obtain the baseline. Then, the PBS buffer was injected into the fluid channel to measure the I-V curve as the blank experiment. After that, different concentrations of IL-6 were injected sequentially to measure the I-V curves. The shifts in the gate voltage were analyzed at the constant drain current of 1´10-9A (indicated by the horizontal red line in Figure R2). The data obtained from one of the devices are shown in Figure R2. In addition, the data recorded from twelve devices (channels) are shown in Figure R1. In order to make the reader understand how we collected these data, we added these two figures (Figure R1 and R2) in Supplementary materials (Figures S5 and S6; mentioned in Section 3.4) and some new descriptions about the stability test and the statistical analysis in Section 2.5.
In Section 2.5: “The stability of the SiNW-FETs without any modification had been tested in the BTP buffer (50 mM, pH7.4), and the Id−Vg curves were measured 20 times and the first Id−Vg curve for each channel was adopted as the baseline (shown in Figure S2). The values of average shift and standard deviation of gate voltage at the drain current (Id) of 1 × 10−9 A are 0.2 mV and 5.5 mV, respectively. The SPSS software (SPSS Inc., Version 18.0, Chicago, USA) was used to perform the statistical analysis. The data obtained from detecting IL-6 with different concentrations by using the device with the mSAMs-GA-Apt layer were examined using one-way analysis of variance (ANOVA) with Tukey's HSD test.”
Figure R1. The Id-Vg curves obtained from twelve channels of the SiNW-FET by using the aptasensor in the detection of IL-6.
Figure R2. The experimental data of using the SiNW-FETs with the aptamer for the detection of IL-6 from one of twelve channels. The changes of gate voltage were analyzed at the constant drain current of 1´10-9A.
1.d) Plotting and calculation of regression line on Fig.5 are seemingly meaningless, because i) the data of Fig.5a is obviously scattered, ii) uncertainties (error bars) of Fig. 5b and 5c are so remarkable, and iii) Langmuir isotherm or Henry’s equation might be better fitted with the data of Fig. 5c.
Response:
According to the reviewer's suggestion, the regression lines in Figure 5 are removed.
1.e) More terribly, the uncertainty shown in Fig. 6 is too significant to validate the detection.
Response:
According to the reviewer's suggestion, we checked the measured data again and discarded some experimental values from the nanowires with lousy quality. For example, the data produced from the nanowire as shown in Figure R3(a) were discarded, and the data acquired from the nanowire with good quality (Figure R3(b)) were taken into calculation (The Id -Vg curves obtained from twelve channels of the SiNW-FET by using the aptasensor are shown in Figure S4.). Then, we use the rest of the data to replot Figure 6. The data presented in the revised Figure 6 have smaller deviation values. By using one-way analysis of variance (ANOVA) with Tukey's HSD test, the statistical results show that some data between adjacent groups reach statistical significance (100 fM vs. 1pM; 100 pM vs. 1 nM). The statements about Figure 6 are corrected in the revised manuscript. A paragraph is added in Discussion to discuss this issue.
“However, the large values of standard deviations shown in Figure 6 might cause measurement uncertainties. The statistical results (Table S4) revealed that the measured signals obtained from the group with the IL-6 concentrations of 1 pM (21 pg/mL) could not have significant differences to its adjacent groups (10 pM and 100 pM). Therefore, using the SiNW-FETs with the mSAMs-GA-Apt layer for distinguishing the IL-6 concentrations in the range between 1 pM and 100 pM could be problematic.”
- (b)
Figure R3. The I-V curves from different nanowires. (a) The nanowire with lousy quality. (b) The nanowire with good quality.
- Please provide characteristic data (e.g., Id-Vg and Id-Vd) of the sensor(s), or a reader cannot tell where the linear region of FET is.
Response:
As per the reviewer's suggestion, the Id-Vg curves of the sensors are provided in the Supplementary materials (Figures S3, S5, and S6).
- To improve the credibility of this work, please consistently provide the product number of each materials/reagent.
Response:
As per the reviewer's suggestion, we add the product number/catalog number of each materials/reagent in the revised manuscript.
- What does the “theoretical” length of the aptamer come from? This is a bold argument (at Line. 241 and 242).
Response:
Thanks for the reviewer's valuable comment. One base pair corresponds to approximately 3.4 Å. For the aptamer with 31 mers, the statement that the aptamer has a theoretical length of 10.54 nm should be wrong because the aptamers themselves can form special secondary structures. The wrong statement was deleted in the revised manuscript. We use the Mfold web server to predict the possible structures of the anti-IL-6 aptamer and give reasonable explanations in the revised manuscript:
“The sizes of aptamers are usually about 3-5 nm [26]. However, the Mfold web server [27] predicts the anti-IL-6 aptamer can have two structures (shown in Figure S3). Because each base pair has a length of 0.34 nm, one of the predicted structures can have a length of up to 8.16 nm. Based on the AFM results, we speculated that the structure shown in Figure S3(b) might be the aptamer structure presented on the sensor surface. The measured height of the immobilized aptamer was also slightly higher than the expected value.”
- And the statement (at Line. 244) contradicts with the measured result (at Line. 238).
Response:
Thanks for carefully reviewing. I correct “less” to “higher” in the statement at Line 244.
Minor comments:
- The paper needs revising by a native English speaker. There are several awkward phrases and typos. Verb tense needs to be addressed and consistent.
Response:
It is a good suggestion. For this time, I check the manuscript thoroughly again, and I try my best to find awkward phrases and typos and correct them. The verb tense is also considered in this revision. If the quality of English still can't fulfill the requirements, I will submit the manuscript to a native English speaker for proofreading.
- In Introduction, the statements (at Line. 32 – 37 on Page.1) regarding COVID-19 testing methods should be revised. An antibody test is used to confirm the past infection, while an antigen test verifies current infection. And RT-PCR is still the gold standard when counting the number of COVID-19 cases.
Response:
Thanks for reviewer's comment. I correct some words in the statements at Line 32 – 37.
- Plus, there is a critical typo at Line. 90 on Page. 2.
Response:
Thanks for carefully reviewing. I correct “antigen” to “aptamer”.

Reviewer 2 Report
The paper explains the detection of interleukin 6 using silicon nanowires FETs and follows other papers in the field of detection using Si nanowires published by the same group. From the reviewer’s point view some information and details are missing and the reader cannot fully appreciate the results without them.
Please find below some technical questions/comments:
- Section 2.3: authors explain the chemical surface treatments, but the reasons of the steps are not always clear (eg, why 2 times ethanol?). No comment is done on the presence of a native oxide on the NW and the fact that it has been removed.
- Same section, the APTES solution is used and in discussions some comments on the concentrations and solvent are given but the authors don’t give any numbers/details on the solution APTES used in their experiments.
- End of section 2.3, authors talk about the electrical measurements and give a drain current value for the shifts’ extractions, but Vd is not given and the geometry of the NW is not given either. We don’t know if the Id value used in the subthreshold region of the FET or above it.
- Authors should show some Id(Vg) curves (at least in the supplementary information)? And they should explain was done in terms of extractions and to what regime corresponds the Id value for example.
- Section 3.2 – 1st phrase not clear to me.
- Bottom page 5, heights are given with 2 digits (eg 9.43nm). Is the last digit significant from the physics point of view?
- Please check the captions of Fig 1 and 2
- AFM data in fig 2, the rectangular graphs are not clearly explained.
- Is the difference in roughness related somehow to the electrical results? To the standard deviations?
- Section 3.3, lines 294-300: authors explain that the shifts in Vg must be 3 times higher than standard deviation. Why 3?
- It is not clear to the reviewer whether the data deltaV versus concertation for a given surface treatment was obtained with different NW or if it is the same NW.
- Could the authors comment on the reproducibility of the response between different NW?
- Authors should give information about the NW used (at least geometry, contact). Where is Vg applied? Is it a backgate? What’s the oxide gate?
- The comments on the linearity of the Fig 5b is not relevant in my opinion since the authors state that the response is regarded as background noise.
- Raw data for FIG 5 should be shown (at least some examples) and the direction of shifts should be commented. Is it in agreement with the charges added on the NW by the surface treatments?
- In Fig 5, you should add the threshold deltaV (the value 3*higher…); this will increase the visibility of your data.
- How did you obtain the error bars in the Fig 5?
- Comments in lines 349-352: were the NW used identical or this could be related to the device-to-device variation of response?
- It is not very clear to the reviewer why the authors talk about the Debye leght.
- I would avoid commenting on the linearity or LOD using data that was explained as under the noise limit.
Author Response
Response to Reviewers
The paper explains the detection of interleukin 6 using silicon nanowires FETs and follows other papers in the field of detection using Si nanowires published by the same group. From the reviewer’s point view some information and details are missing and the reader cannot fully appreciate the results without them.
Please find below some technical questions/comments:
- Section 2.3: authors explain the chemical surface treatments, but the reasons of the steps are not always clear (eg, why 2 times ethanol?). No comment is done on the presence of a native oxide on the NW and the fact that it has been removed.
Response:
Thanks for the reviewer's comment. As this section has declared, the second step of treating the SiNWFET chips with oxygen plasma before surface modification is to form a new oxide layer with the OH groups for further chemical modification. Therefore, even though the native oxide layer, which was contaminated due to exposing to several production steps (packaging, purifying, etc.), was removed, a new oxide layer was formed without being contaminated and consequently benefited from the subsequent surface modification. Therefore, the sentence is revised: “Finally, the oxygen plasma cleaner was utilized again to treat the SiNW-FET device to decompose the surface contaminants and to form a new oxide layer with the OH groups for further chemical modifications.”.
- Same section, the APTES solution is used and in discussions some comments on the concentrations and solvent are given but the authors don’t give any numbers/details on the solution APTES used in their experiments.
Response:
Thanks for the reviewer's comment. I added the information of the APTES solution used in Discussion. Hence, I rewrite the first sentence of Discussion: “Owing to the properties of APTES, the quality of the silica surface modified by the APTES solution with a purity of 99.7% could be influenced by many factors, like temperature, solvent, concentration, reaction time, and moisture [30]”. For reference 31, it is a review article that doesn't provide information about the concentration of APTES.
- End of section 2.3, authors talk about the electrical measurements and give a drain current value for the shifts’ extractions, but Vd is not given and the geometry of the NW is not given either. We don’t know if the Id value used in the subthreshold region of the FET or above it.
Response:
Thanks for the reviewer¢s comment. I provide more detailed descriptions about the measurement in section 2.5. “In the measurement of Id−Vg curve, the drain (Vd) and the source (Vs) voltages were set at 1 and 0.5 volts, respectively. In addition, the gate voltage was swept from 0 V to 2 V with a sweep interval of 0.1 V.”
Since the geometry of the SiNW-FET is not novel and we have used this architecture in several previous publications for many years. So, the geometry of the NW is not provided in the manuscript. The geometry of the NW had been provided in the Supplement Materials of the paper published in Sensors and Actuators B: Chemical (2021, 329, 129150; Supplementary material related to this article can be found, in the online version, at doi: https://doi.org/10.1016/j.snb.2020.129150 ). The figure of NW is shown as the below figure. In this revised manuscript, we provide a sentence describing the NW's geometry in section 2.2.
The added sentence: “Each n-type SiNW-FET device had two nanowires, and each nanowire with a length of 2 mm and a width of 200 nm [20], which had been used in previous reports [20-23].”.
The SEM image of an n-type SiNWFET device, and the nanowires with dimensions of 2-μm length and 200-nm width.
- Vu, C.-A.; Chen, W.-Y.; Yang, Y.-S.; Chan, H.W.-H. Improved Biomarker Quantification of Silicon Nanowire Field-Effect Transistor Immunosensors with Signal Enhancement by RNA Aptamer: Amyloid Beta as a Case Study. Sens Actuators B Chem 2021, 329, 129150.
- Vu, C.-A.; Hu, W.-P.; Yang, Y.-S.; Chan, H.W.-H.; Chen, W.-Y. Signal Enhancement of Silicon Nanowire Field-Effect Transistor Immunosensors by RNA Aptamer. ACS Omega 2019, 4, 14765–14771.
- Chou, W.C.; Hu, W.P.; Yang, Y.S.; Chan, H.W.H.; Chen, W.Y. Neutralized Chimeric DNA Probe for the Improvement of GC-Rich RNA Detection Specificity on the Nanowire Field-Effect Transistor. Sci. Rep. 2019, 9, 11056.
- Hu, W.-P.; Tsai, C.-C.; Yang, Y.-S.; Chan, H.W.-H.; Chen, W.-Y. Synergetic Improvements of Sensitivity and Specificity of Nanowire Field Effect Transistor Gene Chip by Designing Neutralized DNA as Probe. Sci. Rep. 2018, 8, 12598.
- Authors should show some Id(Vg) curves (at least in the supplementary information)? And they should explain was done in terms of extractions and to what regime corresponds the Id value for example.
Response:
Thanks for the reviewer's comment. The SiNW-FET device used in this study was wired and packaged on the printed circuit board (PCB) (shown in Figure S1). The fluid channel was designed and fixed upon the SiNW-FET device to keep the liquid in the nanowire region. The electrical signals from twelve channels could be measured in each experiment by connecting the external electric contacts on the PCB. In this revised manuscript, the Id-Vg curves are provided in figures S3, S5, and S6 of the supplementary materials. Figure S2 is related to the stability test of the SiNW-FET devices. Figure S4 presents the Id-Vg curves obtained from twelve channels of the SiNW-FET by using the aptamer in the detection of IL-6, and Figure S5 shows the enlargement image of channel 2.
- Section 3.2 – 1st phrase not clear to me.
Response:
Thanks for the reviewer's comment. I rewrite this sentence to express the meaning correctly: “According to AFM measurements, the mixed-SAMs had a small Ra value of 0.2 nm, which implied it could form the most uniform layer on the silica surface.”.
- Bottom page 5, heights are given with 2 digits (eg 9.43nm). Is the last digit significant from the physics point of view?
Response:
Thanks for the reviewer's comment. I think the last digit is not significant from the physics point of view. Because the heights of immobilized biomolecules measured by AFM are not the same. Therefore, this sentence is revised follows: The line profiles of AFM images (Figures 2(c) and (d)) for the silicon wafer surfaces with immobilized aptamers indicated that the average height was around 9 nm.
- Please check the captions of Fig 1 and 2
Response:
Thanks for carefully reviewing. I delete the template instructions in the caption of Figure 1. In Figure 2, the caption is revised as: AFM images of silicon wafer surfaces with immobilized antibodies or aptamers.
- AFM data in fig 2, the rectangular graphs are not clearly explained.
Response:
I added a sentence in the legend of Figure 2 to explain the rectangular graph, and the sentence is: The 2D AFM image and line profile of height along the yellow line (the rectangular graph) are shown on the right side of each AFM image.
- Is the difference in roughness related somehow to the electrical results? To the standard deviations?
Response:
In the design of this study, we didn't measure the electric properties of SiNW-FETs after each step of surface functionalization. But, the standard deviations should be related to the fabricated quality of nanowires. Figure S2 shows the stability test of the SiNW-FET devices without any modification. From this figure, some nanowires (channels) have bad output results in the Id-Vg curves, such as channels 7, 10, and 12. I think the primary problem is caused by the consistency of nanowire properties.
- Section 3.3, lines 294-300: authors explain that the shifts in Vg must be 3 times higher than standard deviation. Why 3?
Response:
The limit of detection (LOD) of the instrument has been defined as the average reading of a blank sample (no analyte) plus three standard deviations (SD) of the blank (as shown in Eq. (1)).
The information can refer to the website
(https://sites.chem.utoronto.ca/chemistry/coursenotes/analsci/stats/LimDetect.html) or the chapter of this book (https://www.sciencedirect.com/science/article/pii/B9780128008713000080). For our measurement, the value of blank sample (yblank) was 0 because the baseline was set as the initial point of the signal in each experiment. The obtained LOD value equals to three standard deviations (SD) of the blank experiment.
yLOD = yblank + 3´sblank (1)
- It is not clear to the reviewer whether the data deltaV versus concertation for a given surface treatment was obtained with different NW or if it is the same NW.
Response:.
The SiNW-FET devices was packaged and installed into an electrical circuit board (as shown in Figure S1), and we could obtain signals from 12 channels (devices) (as shown in the below Figure R1 (named as Figure S4 in Supplementary Materials)). We revised Section 2.5 in the manuscript to give more detailed information about the experimental procedures. Different concentrations of IL-6 were injected sequentially to measure the I-V curves. The shifts in the gate voltage were analyzed at the constant drain current of 1´10-9A. Three packaged COB SiNW-FETs were used in each experimental condition, and the experimental conditions were composed of modified layers and biorecognition elements.
Figure R1. The Id-Vg curves obtained from twelve channels of the SiNW-FET by using the aptasensor in the detection of IL-6.
- Could the authors comment on the reproducibility of the response between different NW?
Response:.
Thanks for the reviewer's comment. I think the stability test of the SiNW-FET devices without any modification could be used to answer this question. The test is conducted in the 50 mM BTP buffer (pH 7.4), and the measurement is repeated twenty times. Figure R2 (named as Figure S2 in Supplementary Materials)) shows the results. We added two sentences in Section 2.5: The stability of the SiNW-FET chip had been tested in the BTP buffer (50 mM, pH7.4), and the Id−Vg curves were measured 20 times and the first Id−Vg curve for each channel was adopted as the baseline (shown in Figure S2). From this figure, the curves of some channels exhibited good reproducibility. But, a few nanowires in the chip have lousy quality, like channels 7, 10, and 12. By calculating the data obtained from other channels, the average values of the shift in gate voltage and standard deviation at the drain current (Id) of 1 × 10−9 A are 0.2 mV and 5.5 mV, respectively.
Figure R2. The stability test of the SiNW-FET chip without any modification.
- Authors should give information about the NW used (at least geometry, contact). Where is Vg applied? Is it a backgate? What’s the oxide gate?
Response:
Thanks for the reviewer¢s comment. The geometry and contact of NW have provided in the reply to question 3. The Vg was applied using an electrode which was connected with the electrical board via a red wire, as illustrated in Figure S1. The gate oxide is the oxide layer which has been formed during the oxygen plasma treatment and on the top of the SiNW channels.
- The comments on the linearity of the Fig 5b is not relevant in my opinion since the authors state that the response is regarded as background noise.
Response:.
Thanks for the reviewer's comment. According to your and another reviewer's comments, the regression lines in Figure 5 are removed.
- Raw data for FIG 5 should be shown (at least some examples) and the direction of shifts should be com Is it in agreement with the charges added on the NW by the surface treatments?
Response:.
Thanks for the reviewer's comment. Some examples of raw data for Figure 5 are shown in the following tables, which are presented in Tables S1-S3 of Supplementary Materials. The direction of shift is left, which is the same as the Id-Vg curves shown in Figure S5. As it has been explained in experimental section, the voltage shift values are different between the voltage values before and after detecting the targets. Therefore, the charges added on the NW by the surface treatments do not affect these data.
APTES
|
Groups |
ch1 (mV) |
ch2 (mV) |
ch3 (mV) |
ch4 (mV) |
ch5 (mV) |
|
Blank |
-26.1241 |
-26.8943 |
-10.9187 |
-15.7991 |
-12.75717 |
|
21 pg/mL (1 pM) |
-39.1889 |
-44.3952 |
-18.7305 |
-26.1448 |
-15.3422 |
|
210 ng/mL (10 pM) |
-50.0620 |
-49.4864 |
-21.687 |
-37.2942 |
-18.5127 |
|
2.1 ng/mL (100 pM) |
-48.3429 |
-49.8205 |
-19.9994 |
-34.8226 |
-19.6162 |
|
21 ng/mL (1 nM) |
-50.9766 |
-51.8415 |
-25.7842 |
-35.8480 |
-22.6979 |
|
210 ng/mL (10 nM) |
-51.5569 |
-53.0490 |
-35.9350 |
-48.3660 |
-27.5862 |
APS
|
Groups |
ch1 (mV) |
ch2 (mV) |
ch3 (mV) |
ch4 (mV) |
ch5 (mV) |
|
Blank |
-14.9595 |
-10.5056 |
-6.8901 |
-2.94781 |
-15.7302 |
|
21 pg/mL (1 pM) |
-18.3867 |
-18.4882 |
-10.3212 |
-10.3178 |
-17.6109 |
|
210 ng/mL (10 pM) |
-21.2999 |
-19.5811 |
-17.5142 |
-14.9990 |
-20.6079 |
|
2.1 ng/mL (100 pM) |
-23.2895 |
-24.4332 |
-21.0025 |
-19.5168 |
-24.8676 |
|
21 ng/mL (1 nM) |
-25.8483 |
-26.1696 |
-21.5488 |
-18.8569 |
-29.2589 |
|
210 ng/mL (10 nM) |
-24.9494 |
-23.2263 |
-27.5813 |
-25.6442 |
-29.9680 |
Mixed-SAMs
|
Groups |
ch1 (mV) |
ch2 (mV) |
ch3 (mV) |
ch4 (mV) |
ch5 (mV) |
|
Blank |
-9.8697 |
-15.3840 |
-24.6308 |
-1.9136 |
-1.4856 |
|
21 pg/mL (1 pM) |
-19.5492 |
-22.1176 |
-40.2115 |
-10.5291 |
-6.2969 |
|
210 ng/mL (10 pM) |
-36.3453 |
-26.2015 |
-62.5557 |
-39.6697 |
-16.9319 |
|
2.1 ng/mL (100 pM) |
-46.7271 |
-32.5584 |
-75.8925 |
-76.4369 |
-39.0368 |
|
21 ng/mL (1 nM) |
-61.3980 |
-27.8809 |
-84.6234 |
-92.1436 |
-58.7537 |
|
210 ng/mL (10 nM) |
-63.1347 |
-39.4248 |
-81.3974 |
-95.4557 |
-51.9578 |
- In Fig 5, you should add the threshold deltaV (the value 3*higher…); this will increase the visibility of your data.
Response:.
Thanks for the reviewer's comment. I added two sentences in Section 3.3: The threshold of voltage change could be denoted as ΔVth. If the value of gate voltage change was below the value of ΔVth, it was taken as the background noise. I add values of ΔVth in Figure 5 in the revised manuscript.
- How did you obtain the error bars in the Fig 5?
Response:.
We can obtain signals from 12 channels (devices) in the packaged COB SiNW-FET. After the analysis of Id-Vg curves. The raw data of the gate voltage shift can be obtained, and then we calculate the average and standard deviation values to plot the bar charts with error bars.
- Comments in lines 349-352: were the NW used identical or this could be related to the device-to-device variation of response?
Response:.
The SiNW-FET devices used in this study are all fabricated from the same batch. Although the device-to-device variation still exists among these devices, we perform the same experiments by using three SiNW-FET chips. By using one SiNW-FET chip, we can obtain 12 copies of experimental data of the gate voltage changes corresponding to different IL-6 concentrations in the ideal situation. Therefore, we confirm the SiNW-FET with the use of the anti-IL-6 aptamer in the detection of IL-6 can detect the lower concentration of IL-6.
- It is not very clear to the reviewer why the authors talk about the Debye length.
Response:.
In the Discussion, we mention the Debye length because we employ PEG to prepare hybrid SAMs on the chip surface. I want to highlight the benefits of using PEG SAM on SiNW-FET chips. Therefore, I mention that PEG modification can improve the sensor response, and the Debye length is also discussed here.
19.I would avoid commenting on the linearity or LOD using data that was explained as under the noise limit.
Response:.
Thanks for the reviewer's valuable comment. In this revised manuscript, we avoid using LOD to state the lowest concentration that can be measured. We state that using the anti-IL-6 aptamer in detecting IL-6 with a concentration of 100 pM can be a valid signal. The term LOD is completely removed in the revised manuscript.

Round 2
Reviewer 1 Report
The authors addressed most of the comments, and the missing information has been presented. A minor issue should be fixed: Fig. S1(b) is too blurry. And a more systematic design of experiments is expected to see in the authors' future works.
Overall, quality of the manuscript is significantly improved, an acceptance is suggested.
Author Response
Response to Reviewer 1
Comments and Suggestions for Authors:
The authors addressed most of the comments, and the missing information has been presented.
A minor issue should be fixed: Fig. S1(b) is too blurry. And a more systematic design of experiments is expected to see in the authors' future works.
Overall, quality of the manuscript is significantly improved, an acceptance is suggested.
Response:
Thanks for reviewer's positive comment. I agree that Fig. S1(b) is too blurry. We use a cellphone to take the picture again, and Fig. S1(b) is replaced by the below figure. This figure has a better resolution. In our future works, we will pay special attention to the systematic design of experiments to get more complete and comparable results. We appreciate the reviewer’s valuable comments and suggestions to improve the quality of the manuscript.
